# Study Design, Rationale and Procedures for Human Biomonitoring of Hazardous Chemicals from Foods and Cooking in Korea

**DOI:** 10.3390/ijerph16142583

**Published:** 2019-07-19

**Authors:** Seokwon Lee, Ryoung Me Ahn, Jae Hyoun Kim, Yoon-Deok Han, Jin Heon Lee, Bu-Soon Son, Kyoungho Lee

**Affiliations:** 1Samsung Health Research Institute, Samsung Electronics Co., Ltd. 1 Samsungjeonja-ro, Hwaseong, Gyeonggi-do 18448, Korea; 2Department of Health Sciences, Dongduk Women’s University, 13 gil, 60 Hwarang-ro, Seoul 02748, Korea; 3Department of Environmental Health Science, College of Natural Science, Soonchunhyang University, 22 Soonchunhyang-ro, Asan 31538, Korea; 4Department of Environmental Education, Kongju National University, Gongju 32588, Korea

**Keywords:** biomonitoring, hazardous chemicals, biomarkers, foods and cooking

## Abstract

*Objectives*: A nationwide biomonitoring program identified the long-term trends of environmental exposures to hazardous chemicals in the general population and found geographical locations where body burdens of an exposed group significantly differed from those of the general population. The purpose of this study is to analyze the hazardous compounds associated with foods and cooking in the nationwide general population for evaluation of the environmental exposures and health risk factors and for the establishment of the reference levels at the national level. *Methods*: During 2009–2010, the National Institute of Food and Drug Safety Evaluation (NIFDS) conducted a nationwide human biomonitoring study, including a questionnaire survey and environmental exposure assessments for specific hazardous compounds from foods and cooking among the general population in South Korea. *Results*: A total of 2139 individuals voluntarily participated in 98 survey units in South Korea, including 889 (41.6%) men and 1250 women (58.4%). Bio-specimens (serum and urine) and questionnaires were collected from the study population. Acrylamides, heterocyclic amines (HCAs), phenols, and phthalates were analyzed from urine, and perfluorinated compounds (PFCs) and organic chloride pesticides (OCPs) were analyzed from serum samples. The information on exposure pathway and geographical locations for all participants was collected by questionnaire interviews, which included demographic characteristics, socioeconomic status, history of family diseases, conditions of the indoor and outdoor environment, lifestyles, occupational history, and food and dietary information. *Conclusion*: We describe the design of the study and sampling of human biospecimen procedures including bio-sample repository systems. The resources produced from this nationwide human biomonitoring study and survey will be valuable for use in future biomarkers studies and for the assessment of exposure to hazardous compounds associated with foods and cooking.

## 1. Introduction

Modern people are exposed to thousands of natural and man-made chemicals. The general population can be exposed to hazardous toxic substances through air, water, and foods. There is evidence that environmental exposure to toxic substances in the population of many countries caused various chronic diseases [1,2]. Some countries have created surveillance systems using biomarkers for measurement of exposure to a variety of pollutants including food consumption in the general population from the National Health and Nutrition Examination Surveys (NHANES) in the U.S. [3], the German Environmental Survey (GerES) in Germany [4], and the Korea National Health and Nutrition Examination Surveys in South Korea [5,6]. The potential benefits of the nationwide biomonitoring programs are to identify the long-term trends of environmental exposures to hazardous chemicals in the general population and characterize geographical locations where body burdens of the exposed group significantly differed from those of the general population [7,8]. In Korea, environmental monitoring depended mainly on air, water, food, and soil pollutants. Over the past decade, several initiatives have been launched aiming to include human data in the national environmental framework [6,9,10,11].

In this study, the National Institute of Food and Drug Safety Evaluation (NIFDS) in South Korea planned a nationwide study project of “The collection of biological samples for human biomonitoring and data analysis in 2009–2010”, which promoted biomarker-based exposure assessment of specific pollutants associated with foods and cooking in the general population. The general purpose of the NIFDS project was to establish reference levels among the general population at the national level and to evaluate their variability based on nationwide sample sizes. Second, we wish to develop an association study for research on the sources and routes of hazardous compounds associated with foods and cooking.

This paper describes the research design, rationale, human biosampling procedures, and biospecimen controls for use in the nation-wide biomonitoring program of hazardous chemicals associated with foods and cooking in Korea.

## 2. Study Design

The conceptual basis and design of this study have been made by an expert advisory group, the Environmental Health Study in Korea (EHKor), which included professors, epidemiologists, data analysts, statisticians, and other professionals.

The study design consists of five steps: the selection of subjects, collection of human bio-specimens and questionnaires, sample preparation and delivery, analyses of samples (serum and urine), and establishment of a database system (Figure 1). This study was performed for adults (over 18 years old) for a period of two years (2009–2011), and another similar study was planned for children for a period of two years (2010–2011). 

### 2.1. Selection of Subjects

The target population of this study was adults aged 18–70 years, living in households in Korea. The primary sampling unit for this survey used census enumeration districts, which are defined geographical areas containing around 60 dwellings. Based on data from the 2005 Korea Census, the country was divided into 265,350 enumeration districts. Participants in this study were chosen through a stratified two-stage random sampling. Using the probability proportional to size sampling, 98 enumeration districts were selected for this study. Basically, 10 households were selected by systematic sampling within sample enumeration districts and all adults aged 18–70 years within each household were asked to take part in this survey (Table 1).

The stratification variables for this study were the region and enumeration district classifications, which were determined based on the main house type (residential apartment or single-family house) and urbanization level (town or rural) of the enumeration district. This stratification provided a total of 34 strata for the study. At least 2 enumeration districts were selected within each stratum.

Target subjects of this study were adults (18 to 70 years old). Subjects living in the sample household who agreed to participate in the projects completed informed consent; they were asked to answer the epidemiologic questionnaire in their homes. Biological specimens (serum and urine) were taken at the local public health center by trained phlebotomists. Completed consent participants included 2139 people in 98 sample enumeration districts (Figure 2). The study was approved by the institutional review board at ASAN Medical Center and each subject gave written informed consent (code: 2009-0369).

SAS version 9.4 (SAS Institute Inc., Cary, NC, USA) was used for the statistical analyses in this study. We calculated all estimates using a sampling weight that represented adults aged 18–70 years in Korea. Sampling weights are needed to correct for imperfections in the samples that might lead to bias and other departures between the sample and the reference population. Such imperfections include the selection of units with unequal probabilities, non-coverage of the population, and non-response [12]. The sampling weight calculations in this study account for differential selection probabilities, nonresponse, and post-stratification.

### 2.2. Selection of Hazardous Compounds

In this study, six hazardous chemical compounds, including acrylamides, heterocyclic amines (HCAs), perfluorinated compounds (PFCs), phenols, organic chloride pesticides (OCPs), and phthalates were selected. We analyzed a total of 69 chemicals for human biomonitoring samples collected from serum and urine, as shown in Table 2.

Humans are exposed to acrylamides through food intake, smoking, beverages, cosmetics, etc. According to national research studies of the Food and Drug Administration (FDA), a high concentration of acrylamides has been reported in potato flavored snacks [13]. Although the main pathways of exposure to heterocyclic amines (HCAs) for humans might be surface waters [14], a primary route of HCAs is through food, and a study also showed that there is a positive association between HCA intake and colorectal adenoma risk [15]. In the U.S., serum perfluorooctanoate (PFOA) levels in the randomly selected residents in the water district were significantly higher than those in the general U.S. population, and the main routes of exposure to PFOA were identified as residential drinking water and work during production processes using PFOA [16]. PFCs might be accumulated in the human body [17], and PFC exposure could increase cardiovascular disease and diabetes risks [18]. Phenols are alkyl phenol compounds, which include pentylphenol, octylphenol, and nonylphenol. They are recognized as endocrine disruptors, which interfere with the body’s endocrine system, and produce adverse developmental, reproductive, and neurological problems. They are used mainly as raw materials for cosmetics and perfumes, surfactants, or propellant for pulp and paper mills, and in herbicide sprays or pesticides and toxic agents, polyvinyl chloride (PVC), polyvinyl acetate (PVA), styrene–acrylonitrile ink, anti-oxidants, or stabilizers for synthetic rubber manufacturing. In particular, we have recently made an effort to ensure that food is safe from endocrine-disrupting chemicals in order to protect human health from exposure to bisphenol A, which is a well-known endocrine disruptor found in food containers and packaging [19]. Organic chlorine pesticides (OCPs) are one type of persistent organic pollutant (POP) that are hardly decomposed by chemical, biological, and photochemical reactions in the environments. A number of hazardous chemicals among POPs have been used in pesticides, organic solvents, PVC (polyvinyl chloride), and pharmaceutical materials. They could be generated as potential by-products in the production processes. Di (2-ethylhexyl) phthalate (DEHP), one of the phthalate esters, accounted for over half of the entire usage. DEHP is a colorless, odorless, oily liquid. It is comprised of 40–50% of the weight of the final plastic product [20], and it is banned from use in toothbrushes, rubber nipples, young children’s toys, etc. However, it is allowed for use in food packaging, toys, and medical supplies (serum storage containers and intravenous tubing, etc). Its average use is known to be 4–30 μg/kg per day [21].

### 2.3. Collection of Bio-Specimens

This study involved collection of bio-specimens (serum and urine) from selected subjects for human biological monitoring of harmful substances. During the preparation of the sample collection, sample containers were checked for free of target chemicals. Serum separate tube (SST) was used for serum sampling. Sampling staffs were sufficiently trained using specific protocols, which included the materials necessary for sampling, procedures for taking serum and urine, and a flow diagram from welcome to the final goodbye greeting for visiting subjects in the sampling place. The consented subjects received instructions regarding the appointed place and day/time, fasting of breakfast, and holding of morning urine [22]. The volume of urine necessary from each subject was at least 45 mL for analysis of acrylamides, HCAs, phenols, and phthalates. Analysis of each chemical group required a 5 mL aliquot and backup sample. Urine collection cups were marked on the 50 mL amount of urine for reference while subjects collected urine.

Blood was collected from each subject in the amount of approximately 13–15 mL with two collection vacutainers, for which at least 5.3 mL was required as a serum for analysis of OCPs, PFCs, and total lipid. The collected sample was smoothly mixed at room temperature for 30 min for anti-coagulation using the mixture. The serum was then separated from the blood sample using a centrifuge. Serum was moved to new vacutainers, which were stored in ice boxes for stabilization. During the separation of sample serums, the erythrocytosis serums were no longer used for analysis. All clinical pathology technicians were careful not to make hemolytic serum, and not to have contact with subjects’ blood. All sampling units have an emergency system and program. In cases of pricking by the needle of the syringe during blood sampling procedures, immediately wash the wound with flowing tap water, and transfer to the hospital along with the blood donor in order to investigate whether the technicians were infected or not. Later, report the situation to the headquarters.

Two kinds of labels were used, and bar-coded. One was used for sampling procedures, and another was used for differentiation of the aliquoted samples. Labels provided information on the subject identification number and dates. The labeled sampled containers, including bio-specimens (serum and urine), were wrapped with aluminum foil to break the sunlight, and kept refrigerated at 2–8 °C.

### 2.4. Sample Delivery and Management

The main principles for sample delivery were the maintenance of the cold chain and hand-to-hand system (Figure 1). For cold chain delivery, we manufactured a new delivery icebox, which can maintain the inner temperature to 2–8 °C for 48 h when the outer temperature is 30–35 °C. Our specimen samples were carried from the local sampling unit to the splitting laboratory while maintaining a constant temperature using the refrigerating icebox within 48 h. The icebox was composed of compacted hard plastics; the inner temperature was maintained at 2–8 °C using frozen, portable, insulated ice packs. Upon their arrival at the laboratory, the refrigerated bio-samples were split for the analysis of the aliquots in the refrigeration room. They were then stored at −70 °C until transport to the analyzing laboratory.

A computerized inventory tracking system was developed, so that the storage location of all samples for each participant could be quickly determined. The biological sample repositories for the study are equipped with appropriate alarm systems and emergency electricity backup to prevent accidental thawing. The collected samples in local sampling units were hauled to the person in charge of transportation, in order confirm whether all of the downloaded checklists had been written or not, and the signed checklists were uploaded on cloud storage. The updated checklist was used as raw data for production of bar-coded labels, which were used at the sample splitting laboratory as soon as collected samples arrived; therefore, it was required that the updated checklist be uploaded to cloud storage on the day of delivery. In addition, statistics for the collecting ratio, sex, age, progress, and other information, depending on each professor and region, were posted on cloud storage by utilizing the uploaded data. Sampling time was filled in on the checklist because it was constantly inserted at the time of handling, dispensing, and storage of samples for use in quality management of sampling data.

### 2.5. Sample Analyses and Quality Control

Hazardous compounds were analyzed at six analytical laboratories in each group. Analysis of acrylamides in urine was performed at the chemical analytic laboratory of the Korea Institute of Science and Technology (KIST). Four types of chemicals of acrylamides, which included acrylamide, Glycinamide, AAMA, and GAMA, were analyzed by HPLC (Varian 212, Walnut Creek, CA, USA) with MS/MS detector (Varian, Walnut Creek, CA, USA). Analysis of Heterocyclic amines (HCAs) in urine was performed at the chemical analytic laboratory of the Korea Basic Science Institute (KBSI). Eight metabolites of HCAs, which included IQ, MeIQ, MeIQX, Glu-P-1, Glu-P-2, PhIP, AαC, and MeAαC were analyzed by HPLC (Nanispace Si-2, Shiseido, Japan) with MS detector (LCQ DECA XP, Thermo Finnigan San Jose, CA, USA). Analysis of perfluorinated compounds (PFCs) in serum was performed at the chemical analytic laboratory of Kyunghee University in South Korea. Ten types of chemicals of PFCs, including PFPA, PFHxA, PFHpA, PFOA, PFNA, PFDA, PFHxS, PFHpS, PFOS, and PFNS, were analyzed by HPLC (Agilent 1200 Series CG1367B) with MRM detector (multiple reaction monitoring) of ESI(−)-MS/MS (API 3200, MDS SCIEX, Concord, ON, Canada).

Analysis of phenols in urine was performed at the organic chemical analytic laboratory of KIST. Fifteen metabolites of phenols, which included *t*-BP, *n*-BP, *n*-PP, *n*-HX, *n*-HP, *t*-OP, *n*-OP, NP, bisphenol-A, triclosan, 2.4-dichlorophenol, 2,5-dichlorophenol, 2,4,6-trichlorophenol, and 2,4,6-trichlorophenol were analyzed by separation with GC (Agilent 6890 Series, Santa Clara, USA), and detection with MS (Agilent G1701DA, Santa Clara, USA). Analysis of organic chloride pesticides (OCPs) in lipids of serum was performed at the organic chemical analytic laboratory of KIST. Eighteen metabolites of OCPs, which included hexachlorobenzene, heptachlor, aldrin, oxychlorodane, heptachlor epoxide, *cis*-chlordane, *o,p’*-DDE, *trans*-chlordane, *trans*-nonachlor, *p,p’*-DDE, Dieldrin, *o,p’*-DDD, endrin, *p,p’*-DDD, *o,p’*-DDT, *cis*-nonachlor, *p,p’*-DDT, and mirex were analyzed by separation with GC (Agilent, Palo Alto, CA, USA), and detection with MS (Agilent, Palo Alto, CA, USA). Analysis of phthalates in urine was performed at the chemical analytic laboratory of Eulji University in South Korea. Fifteen metabolites of phthalates, which included MMP, MEP, MnBP, MiBP, MBzP, MCHP, MEHP, MEOHP, MEHHP, 2cx-MMHP, 5cx-MEPP, MnOP, MnPP, MiDP, and MiNP were analyzed by separation with HPLC (Nanospace SI-2, Shiseido, Japan), and detection with MS/MS (LCQ DECA XP, Thermo Finnigan San Jose, CA, USA) (Table 2).

All laboratories verified the analytical methods of all metabolites of hazardous chemicals by enforcing the International Conference on Harmonization of Technical Requirements (ICH) and FDA guidelines [23] during a period of one year (2009). The applied analytic methods were identified and developed to the standard protocols by evaluation of the specificity, linearity, precision, accuracy, sensitivity, and recovery rates during intra-day and inter-day tests. Purified standard materials were used for recovery tests; R squares of linearity rose above 0.995 in the calibration curve with 2, 1.5, 1.0, 0.5, 0.1, 0.01, and 0.001-fold diluted by a standard solution. In relation to the detection limit and the limit of quantification, valid data were applied to be 5 of the S/N ratios, below 20% of precision, and 70~120% of accuracy. During the analysis of samples, quality assurances were checked continually; coefficients of variation (CV) were required to be less than 20%, and analytical procedures were kept with the developed protocols, and so forth.

### 2.6. Questionnaires

The questionnaire for adults was developed as follows: investigation and statistical analysis of the presented literature, selection and structuring of items, development of the preliminary questionnaire, pre-survey with the first questionnaire, correcting the questionnaire, and development of the final questionnaire. After collection, the references/documents of six hazardous compounds (e.g., acrylamides, HCAs, PFCs, phenols, OCPs, phthalates) were investigated for their sources and exposure routes as the preliminary questionnaire items. The items were grouped for their structures according to their similar characteristics and patterns. The preliminary questionnaire was developed with these structuring items, and then used in the pre-survey. With the results of the pre-survey, the final items and structure of the questionnaire were decided, and its phrases were developed for easy comprehension as a questionnaire for general adults. Our final questionnaire was composed of seven categories, including living environment, indoor environment, diseases and medication, lifestyles, diet, day record, and general information. The questionnaire included a total of 40 questions, as shown in Table 3; 7 questions on living environment, 3 questions on disease and medication, 7 questions on lifestyle, 13 questions on diet, 3 questions on day records, and 7 questions related to general information (Table 3).

The phases and structures of our questionnaire have been sophisticated through extensive evaluation, consultation, and recommendation of a variety of environmental health scientists and epidemiologists in South Korea. The questionnaire was accepted by the NIFDS and approved by the Institutional Review Board (IRB) at ASAN Medical Center, and each participant agreed and signed in the written informed consent.

## 3. Results of Biomonitoring Data

Table 4 shows the results of biomonitoring data for six hazardous compounds and chemicals analyzed by urine and serum. The geometric mean (GM) and geometric standard error (GSE) for acrylamides in urine among the study participants were 6.77 μg/g creatinine and 0.240, respectively. As for heterocyclic amines, most of biomonitoring samples were below the limit of detection (LOD). Among perfluorocompounds (10 chemicals) in serum, total GMs were below LOD in PFPA, PFHxA, and PFHpA, and 2.825 ng/mL in PFOA, 0.900 ng/mL in PFNA, 2.159 ng/mL in PFDA, 0.432 ng/mL in PFHxS, 0.047 ng/mL in PFHpS, 10.22 ng/mL in PFOS, and 1.005 ng/mL in PFNS. Among phenols (16 chemicals) in urine, 0.111 μg/g creatinine in n-butylphenol, 0.693 μg/g creatinine in t-butylphenol, 0.284 μg/g creatinine in pentylphenol, 0.558 μg/g creatinine in hexylphenol, 0.632 μg/g creatinine in heptylphenol, 5.770 μg/g creatinine in n-octylphenol, 0.582 μg/g creatinine in t-octylphenol, 3.650 μg/g creatinine in nonylphenol, 1.880 μg/g creatinine in bisphenol-A, 0.139 μg/g creatinine in 2,4-dichlorophenol, 0.418 μg/g creatinine in 2,5-dichlorophenol, 0.081 μg/g creatinine in 2,4,5-trichlorophenol, 0.369 μg/g creatinine in 2,4,6-trichlorophenol, 1.650 μg/g creatinine in triclosan and 4.060 μg/g creatinine in benzophenone-3. Total GMs of OCPs (18 chemicals) in t-lipid of serum were below LOD. Among phthalates (10 chemicals) in urine, total GMs were 41.740 μg/g creatinine in MnHP, 16.970 μg/g creatinine in MiBP, 15.730 μg/g creatinine in MBzP, 0.188 μg/g creatinine in MCHP, 0.057 μg/g creatinine in MnOP, 8.680 μg/g creatinine in MEHP, 17.510 μg/g creatinine in MEOHP, 38.130 μg/g creatinine in MEHHP, 0.070 μg/g creatinine in MiNP, and 0.068 μg/g creatinine in MiDP. All concentrations of biomonitoring samples analyzed in serum and urine showed the reference levels among the study participants selected from the general population in Korea.

## 4. Data Management System

Data obtained from this study were stored in the developed database system. Figure 3 is the entity–relationship diagram (ERD), which was composed of two systems. One was the management system for the data collected during the projects. This included the input subsets of survey data and analytical results of hazardous chemicals, window subset of statistical results of data in real-time, and merged the subset of questionnaire data and analytical results on chemicals. The other was the error checking system. The input system had the automatic function of logic error checking, for which illogical data were alarmed and held to input until the problems were resolved. All data were inputted to the database by qualified field investigators who could access the input system of the website with his/her certificated ID, and data from the questionnaire were inputted twice in order to minimize input errors. After completion of data input in local sampling units, questionnaires were transported to central headquarters, and input again with the same questionnaires to double-check for input errors. Researchers in each analytical laboratory also inputted the resulting data on hazardous chemicals at regular intervals. Local researchers, analytical researchers, and headquarters were able to see the statistical results of input data in real-time, and to effectively manage the national spread sampling units with this system. The data management system can be accessed via a website link of https://www.data.go.kr/.

## 5. Conclusions

The well-designed biomonitoring study was conducted through good sampling and handling, analysis, surveys, etc. To perform an analysis and assessment of hazardous substances, the effectiveness of biomonitoring sampling requires an appropriate strategy of implementation, exact sampling time, standardized analytical procedures, and criteria for interpretation of results [38,39].

The KiFDS project, ‘The collections of biological samples for human biomonitoring and data analysis’, is the first one of the largest nationwide systematic population-based studies on specific biomarkers for pollutants exposed from foods and cooking in South Korea. In this study, we describe the study design, rationale, and methodology, as well as baseline characteristics of the study participants. The study design was stable, reproducible and reliable in terms of management of the nationwide human biomonitoring surveys in South Korea. The human body can be exposed to chemicals analyzed in this study through the food preparation process and/or residual chemicals in foods. A number of chemicals HCA, OCPs, PFCs, phthalates, and phenols were identified at the significantly low levels; this was thought to be related to cooking, containers, environmental contamination, and food consumption.

Therefore, the study results were significantly important and reliable because we identified the national reference levels for hazardous compounds and chemicals from foods and cooking. Further studies are required to identify the significant associations between environmental exposures to food intake and cooking and adverse health outcomes among the general population in Korea in the future.

## Figures and Tables

**Figure 1 ijerph-16-02583-f001:**
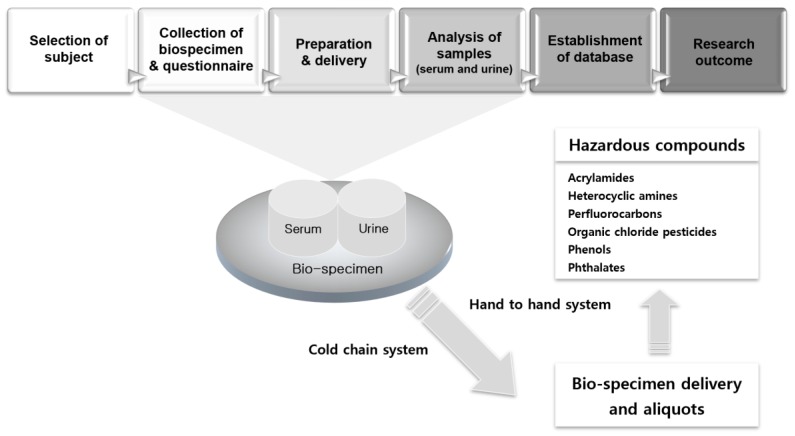
Flow-chart for the human biomonitoring survey.

**Figure 2 ijerph-16-02583-f002:**
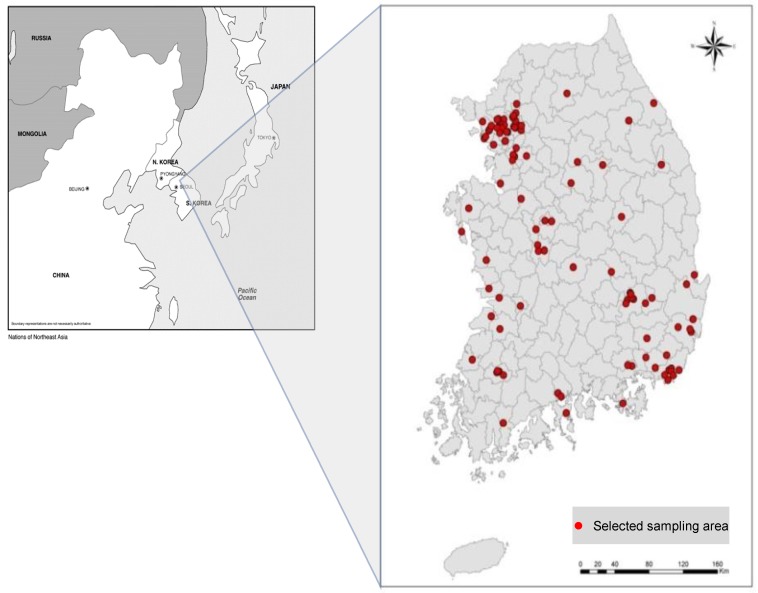
A total of 2139 persons from 98 units participated in the human biomonitoring survey.

**Figure 3 ijerph-16-02583-f003:**
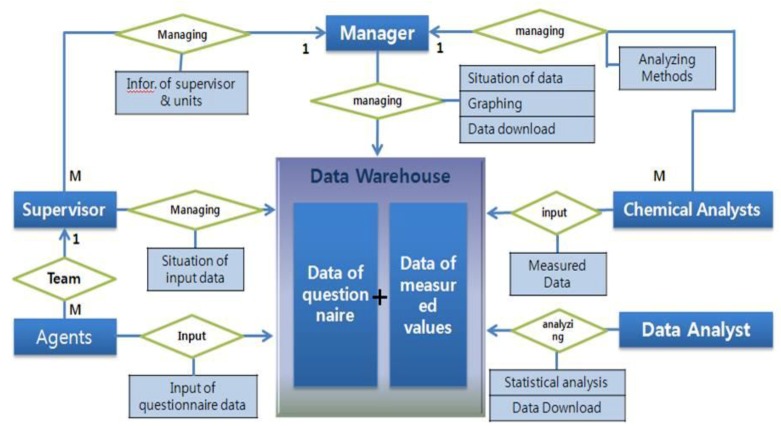
A web-based entity–relationship diagram (ERD) system for the management of questionnaire data and biomonitoring data in this study.

**Table 1 ijerph-16-02583-t001:** Demographic characteristics of the target population participating in this study.

Characteristic	N	%
Total	2139	100
Age		
<29	293	13.7
30–39	474	22.2
40–49	518	24.2
50–59	488	22.8
60–69	366	17.1
Sex		
Male	889	41.6
Female	1250	58.4
Marriage status		
Single	307	14.4
Married, joint/widow	1680	78.6
Married, separate	150	7
Monthly family income		
Less than $1000	466	21.8
$1000–$3000	1010	47.2
$3000–$5000	494	23.1
Over $5000	167	7.9
Residential area		
Urban	1589	74.3
Suburban	64	3.0
Rural	486	22.7

**Table 2 ijerph-16-02583-t002:** Hazardous compounds and their chemicals collected in this study.

Hazardous Compound	Chemical	Abbreviations	Analytical Method
Acrylamides			
	Acrylamide		EPA method 8316
	Glycidamide		
	*N*-acetyl-S-(2-carbamoylethyl)cystein		
Heterocyclic amines			
	2-amino-3-methylimidazo[4,5f]quinoline	IQ	
	2-amino-6-methyldipyrido[1,2-a:3′,2′-d]imidazole	Glu-P-1	
	2-amino-dipyrodo [1,2-a:3′,2′-d]imidazole	Glu-P-2	
	2-amino-3,8-dimethyl-Imidazo[4,5-f] quinoxaline	MeIQx	
	2-amino-3,4-dimethyl-imidazo[4,5-f]quinoline	MeIQ	
	2-amino-1-methyl-6-phenylimidazo[4,5-b]pyridine	PhIP	
	2-amino-9H-pyrodo[2,3-b]indole	AαC	
	2-amino-3-methyl-9H-pyrodo[2,3-b]indole)	MeAαC	
Perfluorocarbons			EPA method 537
	Perflouropentanoic acids,	PFPA	
	Perflourohexanoic acids,	PFHxA	
	Perflouroheptanoic acids,	PFHpA	
	Perfluorooctanoic acids,	PFOA	
	Perfluorononanoic acids,	PFNA	
	Perflourodecanoic acids,	PFDA	
	Perflourohexanesulfonic acids,	PFHxS	
	Perflouroheptanesulfonic acid,	PFHpS	
	Perflourooctanesulfonic acids,	PFOS	
	Perflourononanesulfonic acids)	PFNS	
Phenols			EPA method 604
	t-butylphenol	t-BP	
	n-butylphenol	n-BP	
	n-pentylphenol	n-PP	
	n-hexylphenol	n-HX	
	n-heptylphenol	n-HP	
	t-octylphenol	t-OP	
	n-octylphenol	n-OP	
	Nonylphenol	NP	
	Bisphenol-A		
	Benzophenone-3		
	Triclosan		EPA method 1694
	2,4-dichlorophenol		EPA method 8250
	2,5-dichlorophenol		
	2,4,6-trichlorophenol		
	2,4,5-trichlorophenol		
Organic chloride pesticides (OCPs)			EPA method 8081B
	Hexachlorobenzene		
	Heptachlor		
	Aldrin		
	Oxychlorodane		
	Heptachlor epoxide		
	cis-chlordane		
	o,p’-DDE		
	trans-chlordane		
	trans-nonachlor		
	p,p’-DDE		
	Dieldrin		
	o,p’-DDD		
	Endrin		
	p,p’-DDD		
	o,p’-DDT		
	cis-nonachlor		
	p,p’-DDT		
	Mirex		
Phthalates			EPA method 8061A
	Mono-methyl phthalate (4376-18-5)	MMP	
	Mono-ethyl phthalate (2306-33-4)	MEP	
	Mono-n-butyl phthalate (131-70-4)	MnBP	
	Mono-isobutyl phthalate	MiBP	
	Mono-benzyl phthalate (2528-16-7)	MBzP	
	Mono-cyclohexyl phthalate (7517-36-4)	MCHP	
	Mono-2-ethylhexyl phthalate (4376-20-9)	MEHP	
	Mono-(2-ethyl-5-oxohexyl) phthalate	MEOHP	
	Mono-(2-ethyl-5-hydroxyhexyl) phthalate	MEHHP	
	Mono-(2-carboxy-methyl) hexyl phthalate	2cx-MMHP	
	Mono-(5-carboxy-2-ethyl) pentyl phthalate	5cx-MEPP	
	Mono-n-octyl phthalate (5393-19-1)	MnOP	
	Mono-n-pentyl phthalate	MnPP	
	Mono-isodecyl phthalate	MiDP	
	Mono-isononyl phthalate	MiNP	

**Table 3 ijerph-16-02583-t003:** Exposure assessment of hazardous compounds from the questionnaire.

Categories	Code of Questionnaire	Contents	Hazardous Compounds	References
Outside the residential environment	Q1	Type of residential area	Perfluorinated compounds	[24]
	Q2	Housing type		
	Q3	The contaminated facilities around residents		
	Q4	Vehicles and fuel type		
Indoor	Q5	Interior	Perfluorinated compounds Acrylamides	[24,25,26]
	Q6	Detergent	Perfluorinated compounds	[27,28]
	Q7	Wax and spray	Perfluorinated compounds Organochlorine	[24,29]
Disease and medication	Q8	Medical history		
	Q9	Medicines		
	Q10	Health food		
Lifestyles	Q11	Drinking		
	Q12	Smoking		
	Q13	Exercise		
	Q14	Usual activity		
	Q15	Gore tex use	Perfluorinated compounds	[30]
	Q16	Anti-wrinkle clothes use		
	Q17	Makeup supplies use	Perfluorinated compounds Acrylamides	[13,24]
	Q18	The number of meals per day		
	Q19	Dining area (home or others)		
	Q20	Beverage intake	Organochlorine	[29,31]
	Q21	Food containers	Phenols	[26]
	Q22	Fast foods intake	Acrylamides Phenols	[26,30,32]
	Q23	Cooking methods of meat and fish	Heterocyclic amines Perfluorinated compounds	[13,33]
	Q24	Raw potato storage temperature	Acrylamides	[13]
	Q25	Storage containers for side dish in refrigerator	Phenols	[26]
	Q26	Food packaging wrap	Phenols	[26]
	Q27	Using wrap in a microwave oven	Phenols	[26]
	Q28	Use paper cups	Acrylamides	[34]
	Q29	Use teflon status	Perfluorinated compounds	[24,30]
	Q30	Residue pesticides	Organochlorine	[29]
Record the day	Q31	Meal time just before research interview		
	Q32	Grilled meat and fish intake last 3 days	Heterocyclic amines Perfluorinated compounds	[13,33]
	Q33	Eat seafood last 3 days		
General information	Q34	Height		
	Q35	Weight		
	Q36	Education		
	Q37	Marital status		
	Q38	Monthly average income		
	Q39	Occupations 1	Perfluorinated compounds Acrylamides	[24,30,34,35,36]
	Q40	Occupations 2	Phenols Organochlorine	[29,37]

**Table 4 ijerph-16-02583-t004:** Results of biomonitoring data for hazardous compounds and chemicals (N = 1874).

Media	Hazardouscompounds	Chemicals	Abbreviation	Total			Male			Female		
				N	GM	GSE	N	GM	GSE	N	GM	GSE
Urine	Acrylamides	Acrylamide		1873	6.770	0.240	805	8.140	0.410	1068	5.630	0.250
(unit: ug/g creatinine)		Glycidamide		1873	6.620	0.080	805	6.520	0.170	1068	6.720	0.180
		*N*-acetyl-S-(2-carbamoylethyl)cystein	AAMA	1873	29.880	0.930	805	41.530	1.550	1068	21.450	0.940
	Heterocyclic amines	2-amino-3-methylimidazo[4,5-f]quinoline	IQ	1874	0.110	0.000	805	0.110	0.000	1069	0.110	0.000
		2-amino-6-methyldipyrido[1,2-a:3′,2′-d]imidazole	Glu-P-1	1874	0.125	0.000	805	0.125	0.000	1069	0.125	0.000
		2-amino-dipyrodo[1,2-a:3′,2′-d]imidazole	Glu-P-2	1874	0.140	0.000	805	0.140	0.000	1069	0.140	0.000
		2-amino-3,8-dimethylimidazo[4,5-f] quinoxaline	MeIQx	1874	0.145	0.000	805	0.145	0.000	1069	0.145	0.000
		2-amino-3,4-dimethylimidazo[4,5-f]quinoline	MeIQ	1874	0.146	0.000	805	0.145	0.000	1069	0.146	0.001
		2-amino-1-methyl-6-phenylimidazo[4,5-b]pyridine	PhIP	1874	0.130	0.000	805	0.130	0.000	1069	0.130	0.000
		2-amino-9H-pyrodo[2,3-b]indole	AαC	1874	0.061	0.000	805	0.061	0.000	1069	0.060	0.000
		2-amino-3-methyl-9H-pyrodo[2,3-b]indole)	MeAαC	1874	0.115	0.000	805	0.115	0.000	1069	0.115	0.000
	Perfluorecarbons	Perflouropentanoicacids,	PFPA	1874	0.096	0.001	805	0.096	0.001	1069	0.095	0.001
		Perflourohexanoicacids,	PFHxA	1874	0.055	0.000	805	0.055	0.000	1069	0.055	0.000
		Perflouroheptanoic acids,	PFHpA	1874	0.075	0.000	805	0.075	0.000	1069	0.075	0.000
		Perfluorooctanoic acids,	PFOA	1874	2.825	0.082	805	3.451	0.140	1069	2.309	0.088
		Perfluorononanoic acids,	PFNA	1874	0.900	0.057	805	1.252	0.125	1069	0.645	0.063
		Perflourodecanoic acids,	PFDA	1874	2.159	0.046	805	2.252	0.075	1069	2.068	0.049
		Perflourohexanesulfonic acids,	PFHxS	1874	0.432	0.034	805	0.998	0.107	1069	0.186	0.016
		Perflouroheptanesulfonic acid,	PFHpS	1874	0.047	0.002	805	0.059	0.004	1069	0.038	0.002
		Perflourooctanesulfonic acids,	PFOS	1874	10.220	0.170	805	11.620	0.280	1069	8.990	0.190
		Perflourononanesulfonic acids)	PFNS	1874	1.005	0.058	805	1.017	0.085	1069	0.993	0.071
	Phenols	t-butylphenol	*t*-BP	1874	0.693	0.031	805	0.703	0.039	1069	0.682	0.035
		n-butylphenol	*n*-BP	1874	0.111	0.009	805	0.122	0.013	1069	0.101	0.009
		n-pentylphenol	*n*-PP	1874	0.284	0.019	805	0.290	0.025	1069	0.277	0.021
		n-hexylphenol	*n*-HX	1874	0.558	0.035	805	0.498	0.042	1069	0.625	0.044
		n-heptylphenol	*n*-HP	1874	0.632	0.042	805	0.667	0.058	1069	0.598	0.058
		t-octylphenol	*t*-OP	1874	0.582	0.029	805	0.553	0.037	1069	0.613	0.038
		n-octylphenol	*n*-OP	1874	5.770	0.220	805	5.920	0.340	1069	5.630	0.320
		Nonylphenol	NP	1874	3.650	0.330	805	3.610	0.440	1069	3.680	0.390
		Bisphenol-A		1874	1.880	0.074	805	1.888	0.102	1069	1.871	0.076
		Benzophenone-3		1874	4.060	0.330	805	4.460	0.450	1069	3.690	0.390
		Triclosan		1874	1.650	0.110	805	1.610	0.170	1069	1.690	0.140
		2,4,-dichlorophenol		1874	0.139	0.009	805	0.134	0.012	1069	0.145	0.011
		2,5-dichlorophenol		1874	0.418	0.024	805	0.420	0.034	1069	0.416	0.025
		2,4,6-trichlorophenol		1874	0.369	0.026	805	0.389	0.036	1069	0.349	0.027
		2,4,5-trichlorophenol		1874	0.081	0.003	805	0.080	0.004	1069	0.082	0.004
	Organic chloride pesticides (OCPs)	Hexachlorobenzene		1874	0.009	0.001	805	0.010	0.001	1069	0.008	0.001
		Heptachlor		1874	0.012	0.001	805	0.013	0.001	1069	0.012	0.000
		Aldrin		1874	0.050	0.000	805	0.050	0.000	1069	0.050	0.000
		Oxychlorodane		1874	0.005	0.000	805	0.005	0.000	1069	0.005	0.000
		Heptachlor epoxide		1874	0.010	0.000	805	0.010	0.000	1069	0.010	0.000
		*cis*-chlordane		1874	0.010	0.000	805	0.010	0.000	1069	0.010	0.000
		*o,p’*-DDE		1874	0.010	0.000	805	0.010	0.000	1069	0.010	0.000
		*trans*-chlordane		1874	0.050	0.000	805	0.050	0.000	1069	0.050	0.000
		*trans*-nonachlor		1874	0.012	0.001	805	0.013	0.001	1069	0.012	0.000
		*p,p’*-DDE		1874	0.041	0.003	805	0.043	0.004	1069	0.038	0.003
		Dieldrin		1874	0.005	0.000	805	0.005	0.000	1069	0.005	0.000
		*o,p’*-DDD		1874	0.011	0.000	805	0.011	0.000	1069	0.011	0.000
		Endrin		1874	0.025	0.000	805	0.025	0.000	1069	0.025	0.000
		*p,p’*-DDD		1874	0.005	0.000	805	0.005	0.000	1069	0.005	0.000
		*o,p’*-DDT		1874	0.005	0.000	805	0.005	0.000	1069	0.005	0.000
		*cis*-nonachlor		1874	0.053	0.001	805	0.055	0.002	1069	0.052	0.001
		*p,p’*-DDT		1874	0.052	0.001	805	0.052	0.001	1069	0.052	0.001
		Mirex		1874	0.050	0.000	805	0.050	0.000	1069	0.050	0.000
	Phthalates	Mono-methyl phthalate (4376-18-5)	MMP	1874	-	-	805	-	-	1069	-	-
		Mono-ethyl phthalate (2306-33-4)	MEP	1874	-	-	805	-	-	1069	-	-
		Mono-n-butyl phthalate (131-70-4)	MnBP	1874	41.740	1.070	805	38.700	1.230	1069	45.030	1.320
		Mono-isobutyl phthalate	MiBP	1874	16.970	0.490	805	16.000	0.590	1069	18.010	0.610
		Mono-benzyl phthalate (2528-16-7)	MBzP	1874	15.730	0.710	805	14.010	0.600	1069	17.670	1.090
		Mono-cyclohexyl phthalate (7517-36-4)	MCHP	1874	0.188	0.003	805	0.185	0.004	1069	0.192	0.004
		Mono-2-ethylhexyl phthalate (4376-20-9)	MEHP	1874	8.680	0.280	805	8.060	0.330	1069	9.350	0.370
		Mono-(2-ethyl-5-oxohexyl) phthalate	MEOHP	1874	17.510	0.370	805	15.440	0.390	1069	19.880	0.500
		Mono-(2-ethyl-5-hydroxyhexyl) phthalate	MEHHP	1874	38.130	0.990	805	35.160	1.210	1069	41.370	1.080
		Mono-(2-carboxymethyl)hexyl phthalate	2cx-MMHP	1874	-	-	805	-	-	1069	-	-
		Mono-(5-carboxy-2-ethyl)pentyl phthalate	5cx-MEPP	1874	-	-	805	-	-	1069	-	-
		Mono-n-octyl phthalate (5393-19-1)	MnOP	1874	0.057	0.002	805	0.058	0.002	1069	0.057	0.002
		Mono-n-pentyl phthalate	MnPP	1874	-	-	805	-	-	1069	-	-
		Mono-isodecyl phthalate	MiDP	1874	0.068	0.002	805	0.069	0.003	1069	0.067	0.002
		Mono-isononyl phthalate	MiNP	1874	0.070	0.002	805	0.069	0.003	1069	0.071	0.002
Serum	Perfluorecarbons	Perflouropentanoic acids,	PFPA	1874	0.096	0.001	805	0.180	0.085	1069	0.096	0.000
(unit: ng/mL)		Perflourohexanoic acids,	PFHxA	1874	0.055	0.000	805	0.055	0.000	1069	0.055	0.000
		Perflouroheptanoic acids,	PFHpA	1874	0.075	0.000	805	0.075	0.000	1069	0.075	0.000
		Perfluorooctanoic acids,	PFOA	1874	2.825	0.082	805	3.451	0.140	1069	2.309	0.088
		Perfluorononanoic acids,	PFNA	1874	0.900	0.057	805	1.252	0.125	1069	0.645	0.063
		Perflourodecanoic acids,	PFDA	1874	2.159	0.046	805	2.252	0.075	1069	2.068	0.049
		Perflourohexanesulfonic acids,	PFHxS	1874	0.432	0.034	805	0.998	0.107	1069	0.186	0.016
		Perflouroheptanesulfonic acid,	PFHpS	1874	0.047	0.002	805	0.059	0.004	1069	0.038	0.002
		Perflourooctanesulfonic acids,	PFOS	1874	10.220	0.170	805	11.620	0.280	1069	8.990	0.190
		Perflourononanesulfonic acids	PFNS	1874	1.005	0.058	805	1.017	0.085	1069	0.993	0.071
	Organic chloride pesticides (OCPs)	Hexachlorobenzene		1874	0.009	0.001	805	0.010	0.001	1069	0.008	0.001
		Heptachlor		1874	0.005	0.000	805	0.005	0.000	1069	0.005	0.000
		Aldrin		1874	0.050	0.000	805	0.050	0.000	1069	0.050	0.000
		Oxychlorodane		1874	0.005	0.000	805	0.005	0.000	1069	0.005	0.000
		Heptachlor epoxide		1874	0.010	0.000	805	0.010	0.000	1069	0.010	0.000
		*cis*-chlordane		1874	0.010	0.000	805	0.010	0.000	1069	0.010	0.000
		*o,p’*-DDE		1874	0.010	0.000	805	0.010	0.000	1069	0.010	0.000
		*trans*-chlordane		1874	0.050	0.000	805	0.050	0.000	1069	0.050	0.000
		*trans*-nonachlor		1874	0.012	0.001	805	0.013	0.001	1069	0.012	0.000
		*p,p’*-DDE		1874	0.041	0.003	805	0.043	0.004	1069	0.038	0.003
		Dieldrin		1874	0.005	0.000	805	0.005	0.000	1069	0.005	0.000
		*o,p’*-DDD		1874	0.011	0.000	805	0.011	0.000	1069	0.011	0.000
		Endrin		1874	0.025	0.000	805	0.025	0.000	1069	0.025	0.000
		*p,p’*-DDD		1874	0.005	0.000	805	0.005	0.000	1069	0.005	0.000
		*o,p’*-DDT		1874	0.005	0.000	805	0.005	0.000	1069	0.005	0.000
		*cis*-nonachlor		1874	0.053	0.001	805	0.055	0.002	1069	0.052	0.001
		*p,p’*-DDT		1874	0.052	0.001	805	0.052	0.001	1069	0.052	0.001
		Mirex		1874	0.050	0.000	805	0.050	0.000	1069	0.050	0.000

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
