# Peer review of "Study Design, Rationale and Procedures for Human Biomonitoring of Hazardous Chemicals from Foods and Cooking in Korea"

_ijerph, 2019, doi:10.3390/ijerph16142583_

Reviewer 1 Report

The paper titled “Study Design and Procedures for Human Biomonitoring of Hazardous Chemicals from Foods and Cooking” by Seokwon Lee et al., describes “the design of study and sampling of human biospecimen procedures including biosample repository systems” from a “nationwide human biomonitoring study, which included a survey for implementation of an exposure assessment to population to specific harmful materials from foods and cooking”.

The paper is well written, the contents are appropriate and can provide a roadmap for further studies. However, this is a protocol from a human biomonitoring study that was held in 2009-2010. I express my concern regarding the novelty of the publication. At least it should be highlighted the pertinence of the publication after 10 years of the study. I also make some comments that could improve the quality of the paper.

Comment 1: In Abstract the authors start right away with the objective. However, I believe that a small introduction should also be included.

Comment 2: This is a protocol from a human biomonitoring study that was held in 2009-2010. What was the reasoning to submit for publication a protocol with 10 years? I suggest to the authors to highlight the pertinence of this publication, namely with reference to published work coming from this biomonitoring study (also valid in the summary and conclusions section). Another pertinent indication to include may be the periodicity, ie whether there will be a periodic application of biomonitoring or if it was unique. This issue of the novelty should be clarified and discussed.

Comment 3: Also in this context, the majority of the references are not recent. I suggest that more recent references should also be included.

Comment 4: In Figure 1, the authors represent the flow-chart for the human biomonitoring survey. In the Analysis of samples there is a reference to Heavy metals. However, is no mention of them throughout the article. Were they also analysed but not included? This should be clarified and discussed.

Comment 5: In page 6 and 7 the authors write about the EDCs. Indeed, this mention is very important, however one gets the idea that only phenols are EDCs, when OCPs and phthalates are also. This should be clarified.

Comment 6: In page 6, line 167 and 171 the authors write about the urine sampling. However, it is not clear if the collection is the 1st of the morning or a spot at any time (at the time of the appointment). In the same context, I suggest that the authors improve the description or use a schematic of the collection, especially to be clear when and under what conditions (fasting, 1st morning urine? ...).

Minor revisions:

- In page 2, line 52 the authors say “Modern people are exposed to thousands of natural and artificial chemicals every day. However, the general population are exposed to hazardus toxic substances through air, water and food.”. Is “However” well applied? This should be confirmed. Also, it should be “hazardous”.

- In page 2, line 55 to 57 the sentence is a bit confusing. This should be confirmed.

- In page 2, line 59 in the sentence “…to variety pollutants including food…” it should be “…to a variety of pollutants including food…”

- In page 6, line 134 the authors say “These compounds can be found in raw meat from cooked food...”. Is “raw” well applied? This should be confirmed.

- In page 6, line 139 the authors say “…informed as endocrine disruptors...”. Is “informed” well applied? Could it be for example “recognised”. This should be confirmed.

- In page 7, line 160 the authors describe a method to extract phthalates from plastic. Is there any method reference? Also, is “In particles” well applied in the beginning of the sentence?

- The acrylamides nomenclature should be uniform, since it appears with a space or without it.

Author Response

Reviewer’s Comments and Responses

General Comments:

The paper titled “Study Design and Procedures for Human Biomonitoring of Hazardous Chemicals from Foods and Cooking” by Seokwon Lee et al., describes “the design of study and sampling of human biospecimen procedures including biosample repository systems” from a “nationwide human biomonitoring study, which included a survey for implementation of an exposure assessment to population to specific harmful materials from foods and cooking”.

The paper is well written, the contents are appropriate and can provide a roadmap for further studies. However, this is a protocol from a human biomonitoring study that was held in 2009-2010. I express my concern regarding the novelty of the publication. At least it should be highlighted the pertinence of the publication after 10 years of the study. I also make some comments that could improve the quality of the paper.

Specific comments for major revisions:

Comment 1: In Abstract the authors start right away with the objective. However, I believe that a small introduction should also be included.

Response: Yes, you are correct. We added a paragraph in Abstract. Please see below.

Page 1 Line 24-27:

The nationwide   biomonitoring program enables to identify the long-term trends of   environmental exposures to hazardous chemicals in the general population and   to find geographical locations where body burdens of an exposed group   significantly differed from those of the general population. The purpose   of this study is …

Comment 2: This is a protocol from a human biomonitoring study that was held in 2009-2010. What was the reasoning to submit for publication a protocol with 10 years? I suggest to the authors to highlight the pertinence of this publication, namely with reference to published work coming from this biomonitoring study (also valid in the summary and conclusions section). Another pertinent indication to include may be the periodicity, ie whether there will be a periodic application of biomonitoring or if it was unique. This issue of the novelty should be clarified and discussed.

Response: This study was the first nationwide biomonitoring study to collect bio-specimens from large-scale study participants among the general population in Korea a decade ago, but no peer-reviewed article has been published yet since the study was completed. We authors decided to submit this manuscript for introduction of the study design, rationale and procedures of biomonitoring conducted in Korea. This is a reason why we submit this manuscript for the publication. We added paragraphs in Summary and Conclusion sections. Please see below.

Page 14 Line 293-313:

Results of bio-monitoring data

Table 4   showed the results of biomonitoring data for 6 hazardous compounds and chemicals   analyzed by urine and serum. The geometric mean (GM) and geometric standard   error (GSE) for acrylamides in urine among the study participants was 6.77   μg/g creatinine and 0.240, respectively. As for heterocyclic amines, most of   biomonitoring samples were below limit of detection (LOD). Among   perfluorocompounds (10 chemicals) in serum, total GMs were below LOD in PFPA,   PFHxA and PFHpA, and 2.825 ng/ml in PFOA, 0.900 ng/ml in PFNA, 2.159 ng/ml in   PFDA, 0.432 ng/ml in PFHxS, 0.047 ng/ml in PFHpS, 10.22 ng/ml in PFOS, and   1.005 ng/ml in PFNS. Among phenols (16 chemicals) in urine, 0.111 μg/g   creatinine in n-butylphenol, 0.693 μg/g creatinine in t-butylphenol, 0.284   μg/g creatinine in pentylphenol, 0.558 μg/g creatinine in hexylphenol, 0.632   μg/g creatinine in heptylphenol, 5.770 μg/g creatinine in n-octylphenol,   0.582 μg/g creatinine in t-octylphenol, 3.650 μg/g creatinine in nonylphenol,   1.880 μg/g creatinine in bisphenol-A, 0.139 μg/g creatinine in   2,4-dichlorophenol, 0.418 μg/g creatinine in 2,5-dichlorophenol, 0.081 μg/g   creatinine in 2,4,5-trichlorophenol, 0.369 μg/g creatinine in   2,4,6-trichlorophenol, 1.650 μg/g creatinine in triclosan and 4.060 μg/g   creatinine in benzophenone-3. The total GMs of OCPs(18 chemicals) in t-lipid   of serum were below LOD. Among phthalates (10 chemicals) in urine, total GMs   were 41.740 μg/g creatinine in MnHP, 16.970 μg/g creatinine in MiBP, 15.730   μg/g creatinine in MBzP, 0.188 μg/g creatinine in MCHP, 0.057 μg/g creatinine   in MnOP, 8.680 μg/g creatinine in MEHP, 17.510 μg/g creatinine in MEOHP,   38.130 μg/g creatinine in MEHHP, 0.070 μg/g creatinine in MiNP, and 0.068   μg/g creatinine in MiDP. All concentrations of biomonitoring analyzed in   serum and urine showed the reference levels among the study participants   selected from the general population in Korea.

Comment 3: Also in this context, the majority of the references are not recent. I suggest that more recent references should also be included.

Response: Yes, you are correct. We added new references recently published within 5 years. Please see References in our revised manuscript.

Comment 4: In Figure 1, the authors represent the flow-chart for the human biomonitoring survey. In the Analysis of samples there is a reference to Heavy metals. However, is no mention of them throughout the article. Were they also analyzed but not included? This should be clarified and discussed.

Response: Yes, you are correct. We did not analyze heavy metals in this study, thus deleted in Figure 1. Please see the revised Figure 1.

Comment 5: In page 6 and 7 the authors write about the EDCs. Indeed, this mention is very important, however one gets the idea that only phenols are EDCs, when OCPs and phthalates are also. This should be clarified.

Response: Yes, you are correct. We mentioned only Phenol and OCPs as the representative hazardous chemicals, which could possibly be exposed from food intake and cooking. They were well-known hazardous chemicals among EDCs and POPs, respectively. We reviewed several peer-reviewed articles regarding Phenols and OCPs in that paragraphs.

Comment 6: In page 6, line 167 and 171 the authors write about the urine sampling. However, it is not clear if the collection is the 1st of the morning or a spot at any time (at the time of the appointment). In the same context, I suggest that the authors improve the description or use a schematic of the collection, especially to be clear when and under what conditions (fasting, 1st morning urine? ...).

Response: Yes, you are correct. According to several references (Wallner-Liebmann et al. 2015), we collected the first urine in the morning.

Minor revisions:

1. In page 2, line 52 the authors say “Modern people are exposed to thousands of natural and artificial chemicals every day. However, the general population are exposed to hazardus toxic substances through air, water and food.”. Is “However” well applied? This should be confirmed. Also, it should be “hazardous”.

Response: Yes, you are correct. We modified the sentence. Please see below.

Page 2 Line 55-56:

Modern people are exposed to thousands of   natural and man-made chemicals. The general population can be especially   exposed to hazardous toxic substances through air, water and foods.

2. In page 2, line 55 to 57 the sentence is a bit confusing. This should be confirmed.

Response: Yes, you are correct. We deleted the sentence. Please see below.

Page 2 Line 55-57:

In addition, the naturally occurring or man-made   chemicals are present in the atmosphere, food and water, in the workplace and   in many consumer goods. these chemicals can be very complex occuring diseases   in animals, plants and humans from a varuety of potential sources. (deleted)

3. In page 2, line 59 in the sentence “…to variety pollutants including food…” it should be “…to a variety of pollutants including food…”

Response: Yes, you are correct. We modified the sentence. Please see below.

Page 2 Line 58-61:

Some countries have done surveillance systems   using biomarkers for measurement of exposure to a variety of   pollutants including food consumption in the general population from the   National Health and Nutrition Examination Surveys (NHANES) in the U.S. [4], …

4. In page 6, line 134 the authors say “These compounds can be found in raw meat from cooked food...”. Is “raw” well applied? This should be confirmed.

Response: Yes, you are correct. We modified the sentence. Please see below.

Page 2 Line 148-149:

These hazardous compounds can be also found in raw   (deleted) meats from cooked food, …

5. In page 6, line 139 the authors say “…informed as endocrine disruptors...”. Is “informed” well applied? Could it be for example “recognised”. This should be confirmed.

Response: Yes, you are correct. We modified the sentence. Please see below.

Page 2 Line153-155:

They are recognized as endocrine   disruptors, which interfere with the body’s endocrine system, and produce   adverse developmental, reproductive, and neurological problems.

6. In page 7, line 160 the authors describe a method to extract phthalates from plastic. Is there any method reference? Also, is “In particles” well applied in the beginning of the sentence?

Response: Yes, you are correct. We deleted the sentence. Please see below.

Page 8 Line 174-175:

During preparation of the sample collection,   sample containers were checked for free of target chemicals. In particles,   phthalates could be extracted from plastics that are usually used as sample   containers. Ethylene diamine tetra acetic acid (EDTA) vacutainers were used   for blood sampling. (deleted)

7. The acrylamides nomenclature should be uniform, since it appears with a space or without it.

Response: Yes, you are correct. We unified “acrylamides” in all sentences. Please see below.

Page 5 Line 132:

For this   study, six hazardous chemical groups, including acrylamides, …

Page 7 Line 143-146:

Humans are exposed to acrylamides through   food intake, smoking, beverages, cosmetics, etc. According to national   research studies of the Food and Drug Administration (FDA), a high   concentration of acrylamides has been reported in potato flavored   snacks [10].

Page 8 Line 180-181:

The volume of urine necessary from each subject   was at least 45 ml for analysis of acrylamides, HCAs, phenols, and   phthalates.

Page 9 Line 225-229:

Hazardous chemicals groups were analyzed at 6   analytical laboratories in each group. Analysis of acrylamides in   urine was performed at the chemical analytic laboratory of the Korea   Institute of Science and Technology (KIST). Four metabolites of acrylamides,   which included acrylamide, glycid amide, AAMA, and GAMA were analyzed   by separation with HPLC (Varian 212, Walnut Creek, California, USA), and   detection with MS/MS (Varian, Walnut Creek, California, USA).

Page 9 Line 269-271:

After collection, the references/documents of 6   hazardous chemical groups (acrylamides, HCAs, PFCs, phenols, OCPs, phthalates)   were investigated for their sources and exposure routes, which were used as   preliminary items.

- End -

Reviewer 2 Report

The paper describes the experimental design concerning a survey, through humans, of a set of hazardous organic pollutants with many diffusion routes, including food and cooking. The subject is interesting, but the article does not present any data on the survey. This is a strong limit. The authors must explain why this limitation (perhaps they are preparing another MS) but they must avoid any conclusion on data that are not shown.

My suggestions are to change conclusion and, above all, to revise the English that is poor and in many cases, confuse so that the concepts are obscure.

Author Response

Reviewer’s Comments and Responses

Reviewer #2

1. The paper describes the experimental design concerning a survey, through humans, of a set of hazardous organic pollutants with many diffusion routes, including food and cooking. The subject is interesting, but the article does not present any data on the survey. This is a strong limit. The authors must explain why this limitation (perhaps they are preparing another MS) but they must avoid any conclusion on data that are not shown.

Response: Yes, you are correct. We added the demographic characteristics of population in Table 1 and results of biomonitoring data for hazardous compounds and chemicals in Table 4. This study was the first nationwide biomonitoring study to collect bio-specimens from large-scale study participants among the general population in Korea a decade ago, but no peer-reviewed article has been published yet since the study was completed.

2. My suggestions are to change conclusion and, above all, to revise the English that is poor and in many cases, confuse so that the concepts are obscure.

Response: Yes, we modified the results and conclusion. Please see below.

Page 13 Line 293-313:

Results of biomonitoring data

Table 4   showed the results of biomonitoring data for 6 hazardous compounds and   chemicals analyzed by urine and serum. The geometric mean (GM) and geometric   standard error (GSE) for acrylamides in urine among the study participants   was 6.77 μg/g creatinine and 0.240, respectively. As for heterocyclic amines,   most of biomonitoring samples were below limit of detection (LOD). Among   perfluorocompounds (10 chemicals) in serum, total GMs were below LOD in PFPA,   PFHxA and PFHpA, and 2.825 ng/ml in PFOA, 0.900 ng/ml in PFNA, 2.159 ng/ml in   PFDA, 0.432 ng/ml in PFHxS, 0.047 ng/ml in PFHpS, 10.22 ng/ml in PFOS, and   1.005 ng/ml in PFNS. Among phenols (16 chemicals) in urine, 0.111 μg/g   creatinine in n-butylphenol, 0.693 μg/g creatinine in t-butylphenol, 0.284   μg/g creatinine in pentylphenol, 0.558 μg/g creatinine in hexylphenol, 0.632   μg/g creatinine in heptylphenol, 5.770 μg/g creatinine in n-octylphenol,   0.582 μg/g creatinine in t-octylphenol, 3.650 μg/g creatinine in nonylphenol,   1.880 μg/g creatinine in bisphenol-A, 0.139 μg/g creatinine in   2,4-dichlorophenol, 0.418 μg/g creatinine in 2,5-dichlorophenol, 0.081 μg/g   creatinine in 2,4,5-trichlorophenol, 0.369 μg/g creatinine in   2,4,6-trichlorophenol, 1.650 μg/g creatinine in triclosan and 4.060 μg/g   creatinine in benzophenone-3. The total GMs of OCPs(18 chemicals) in t-lipid   of serum were below LOD. Among phthalates (10 chemicals) in urine, total GMs   were 41.740 μg/g creatinine in MnHP, 16.970 μg/g creatinine in MiBP, 15.730   μg/g creatinine in MBzP, 0.188 μg/g creatinine in MCHP, 0.057 μg/g creatinine   in MnOP, 8.680 μg/g creatinine in MEHP, 17.510 μg/g creatinine in MEOHP,   38.130 μg/g creatinine in MEHHP, 0.070 μg/g creatinine in MiNP, and 0.068   μg/g creatinine in MiDP. All concentrations of biomonitoring analyzed in serum   and urine showed the reference levels among the study participants selected   from the general population in Korea.

- End -

Reviewer 3 Report

Comments to the manuscript ijerph-534532. Title: "Study Design and Procedures for Human Biomonitoring of Hazardous Chemicals from Foods and Cooking" for the International Journal of Environmental Research and Public Health.

This manuscript is about the design of the pilot study and its procedures in human biomonitoring of exposure of inhabitants from South Korea to 6 selected hazardous chemicals narrowly related to foods and cooking. In my opinion, this manuscript clearly shows all steps given from the first idea until the end of the process, which a laborious development of every subject implied. I consider that the 6 families of hazardous compounds are well selected and inside every family, they are representative of them. Besides, the biospecimens are well established, taking into account that easiness to get samples and they are almost painless.

Authors have centred this study in chemicals that can be analysed using two main instrumental techniques like HPLC and GC, with MS as a detector in different options. I consider that the description along the manuscript is correct, what is expected for this kind of manuscripts. Anyway, I have some comments to do.

·         The first one is about the bio-specimens. In line 38 blood and urine are the two bio-specimens. According to the Study design, in line 84, serum, blood and urine were the matrices for analysis. However, in figure 1, the authors only talk about blood and urine. The same for lines 107-108 or line 158, but in figure 1 there is a part with an indication of serum and urine bio-specimen. Besides, in line 40, only PFCs were analysed from the blood. Please, check this information to make it consistent along of the text.

In that figure 1, authors also indicate that the analysis of heavy metals is planned or, at least, that is what I understand. I guess it will be a new step for future studies.

·         When the authors describe the hazardous materials, they group them into 6 families. However, the number of selected compounds is lower than the sum of all of them described in table 1. According to this table, they are 69 instead of 65.

·         In lines 134 and 135, there is a bit confused about the description of HCAs and PFCs. I consider that this part of the text is not well delimited.

·         In the section Collection of bio-specimens, the authors say that urine samples were collected with the help of some plastic gloves. I guess that the authors think of the possibilities of cross-contamination with phthalates, although nothing too different is possible to do.

·         I am ignorant about the style of life in South Korea, but I suppose that the comment in line 165-169 is related to the fact of getting better control of the sampling.

·         The description of sample analyses and quality control is very appropriate. However, could the authors add a reference for the analytical method employed in every family of compounds?

·         I have tried to go to the link of the data management system (line 298) and I was unable to enter. Please, check it.

·         Some of the compounds analysed in this study are very persistent, as OCPs are. Mainly thinking of them, they can come from years earlier than the current location of their houses. I also think of the difficulties to separate the possible origin of the pollutants at home or outside the home. Have you asked for the number of hours outside the home or something similar to evaluate it?

·         In the Summary and conclusions section, the authors establish that ‘the results were very meaningful and identified the reference value for these chemicals…’ We are not able to evaluate this comment because we do not have those results.

·         These are my last questions. When do the authors think that the results will be published?  I have been surprised by the only result appeared here that is the ratio of men and women (41.6% and 58.4%, respectively). How many people did not want to participate when they were asked for it in a previous step? Have you thought about a periodical sampling study? Every 5 years, for instance? Very interesting would be to create cohorts, mainly with the 2 years old children to follow in the future. Have you tried to create a biobank with the samples obtained?

·         Please, pay attention to the references and use the abbreviated journal name when it is in a long way form: references 1, 4, 6, 11, 12, 13, 14, 18, 19, 22, 24, 25, 26, 27, 30, and 31.

Author Response

Reviewer’s Comments and Responses

Reviewer #3

This manuscript is about the design of the pilot study and its procedures in human biomonitoring of exposure of inhabitants from South Korea to 6 selected hazardous chemicals narrowly related to foods and cooking. In my opinion, this manuscript clearly shows all steps given from the first idea until the end of the process, which a laborious development of every subject implied. I consider that the 6 families of hazardous compounds are well selected and inside every family, they are representative of them. Besides, the biospecimens are well established, taking into account that easiness to get samples and they are almost painless.

Authors have centred this study in chemicals that can be analysed using two main instrumental techniques like HPLC and GC, with MS as a detector in different options. I consider that the description along the manuscript is correct, what is expected for this kind of manuscripts. Anyway, I have some comments to do.

1. The first one is about the bio-specimens. In line 38 blood and urine are the two bio-specimens. According to the Study design, in line 84, serum, blood and urine were the matrices for analysis. However, in figure 1, the authors only talk about blood and urine. The same for lines 107-108 or line 158, but in figure 1 there is a part with an indication of serum and urine bio-specimen. Besides, in line 40, only PFCs were analysed from the blood. Please, check this information to make it consistent along of the text. In that figure 1, authors also indicate that the analysis of heavy metals is planned or, at least, that is what I understand. I guess it will be a new step for future studies.

Response: Yes, you are correct. We collected biomonitoring samples from serum and urine. PFCs were analyzed from serum. In Fiture 1, we did not analyze heavy metals in this study. We modified all related paragraphs in our manuscript. Please see below.

Page 2 Line 39-40:

Bio-specimens (serum and urine) and   questionnaires were collected from the study population.

Page 8 Line 175-176:

Serum Separate Tube (SST) was used for serum   sampling.

Page 8 Line 176-178:

Sampling staffs were sufficiently trained using   specific protocols, which included the materials necessary for sampling,   procedures for taking serum and urine, and a flow diagram from welcome   to the final good-bye greeting for visiting subjects in the sampling place.

Page 8 Line 186-188:

The collected sample was smoothly mixed   at room temperature for 30 minutes for anti-coagulation using the mixture. Then   serum was separated from blood sample using a centrifuge. Serum   was moved to new vacutainers, which were stored in ice boxes for   stabilization.

2. When the authors describe the hazardous materials, they group them into 6 families. However, the number of selected compounds is lower than the sum of all of them described in table 1. According to this table, they are 69 instead of 65.

Response: Yes, you are correct. We counted the number of hazardous chemicals in Table 2, and they were 69 chemicals. We modified the sentence as you commented. Please see below.

Page 5 Line 132-133:

In this study, six hazardous chemical compounds,   including acrylamides, heterocyclic amines (HCAs), perfluorinated compounds   (PFCs), phenols, organic chloride pesticides (OCPs), and phthalates were   selected. We analyzed a total of 69 chemicals for human biomonitoring samples   collected from serum and urine, as shown in Table 2.

3. In lines 134 and 135, there is a bit confused about the description of HCAs and PFCs. I consider that this part of the text is not well delimited.

Response: Yes, you are correct. We modified the sentence. Please see below.

Page 7 Line 145-152:

Although the main pathways of exposure to   heterocyclic amines (HCAs) for humans might be surface waters [11], a primary   route of HCAs is through food, and a study also showed that there is a   positive association between HCA intake and colorectal adenoma risk [49].   These hazardous compounds can be also found in meats from cooked food, and   the main route of PFCs is also drinking water and occupational exposure [12],   but possibly other items in the living environment, such as kitchen utensils,   foods, shampoo containers, etc. PFCs might be accumulated in the human body   [13], and PFC exposure could increase cardiovascular disease and diabetes   risks [50].

4. In the section Collection of bio-specimens, the authors say that urine samples were collected with the help of some plastic gloves. I guess that the authors think of the possibilities of cross-contamination with phthalates, although nothing too different is possible to do.

Response: Yes, you are correct. We deleted the sentence. There was no cross-contamination because phthalate-free polyethylene gloves were used during the collection of bio-specimens. Please see below.

Page 7 Line 175:

Plastic gloves were distributed for subjects’   urine sampling. (deleted)

5. I am ignorant about the style of life in South Korea, but I suppose that the comment in line 165-169 is related to the fact of getting better control of the sampling.

Response: Yes, you are correct. We deleted the sentences. Please see below.

Page 7 Line 165-169:

Criteria for selection of sampling sites in the   sample survey unit included a location that could be easily accessed by the   subjects and stand by, sufficient space for collection of bio-specimens and   survey questionnaires, and with easy access to toilets for urine sampling.   The best locations were the community health center, the nearby local   hospital, the community service center, administrative offices, or senior   citizen offices. (deleted)

6. The description of sample analyses and quality control is very appropriate. However, could the authors add a reference for the analytical method employed in every family of compounds?

Response: Yes, you are correct. We added references for the analytical methods employed in this study. Please see Table 2.

7. I have tried to go to the link of the data management system (line 298) and I was unable to enter. Please, check it.

Response: Yes, you are correct. We checked and revised a website link of the data management system operated by the Korean government. Please see below.

Page 14 Line 330-331:

The data management system can be accessed via a   website link of https://www.data.go.kr/.

8. Some of the compounds analysed in this study are very persistent, as OCPs are. Mainly thinking of them, they can come from years earlier than the current location of their houses. I also think of the difficulties to separate the possible origin of the pollutants at home or outside the home. Have you asked for the number of hours outside the home or something similar to evaluate it?

Response: Yes, you are correct. We asked whether or not the study participants perform indoor and outdoor exercises regularly over 30 min in the questionnaire surveys. We also asked the geographical location and characteristics in the outdoor environment near residential housings. Please see the questionnaire in Table 3.

9. In the Summary and conclusions section, the authors establish that ‘the results were very meaningful and identified the reference value for these chemicals…’ We are not able to evaluate this comment because we do not have those results.

Response: Yes, you are correct. We added the demographic characteristics of population in Table 1 and results of biomonitoring data for hazardous compounds and chemicals in Table 4. Please see those Tables.

10. These are my last questions. When do the authors think that the results will be published? I have been surprised by the only result appeared here that is the ratio of men and women (41.6% and 58.4%, respectively). How many people did not want to participate when they were asked for it in a previous step? Have you thought about a periodical sampling study? Every 5 years, for instance? Very interesting would be to create cohorts, mainly with the 2 years old children to follow in the future. Have you tried to create a biobank with the samples obtained?

Response: We added the demographic characteristics of study participants in Table 1. But individuals who did not agree to participate in this study were excluded. The Environmental Health studies in Korea (EHKor) have been conducted by the Ministry of Environment (MoE) every 4 years, and the biomonitoring data is regularly collected and stored in the system. A large-scale biobank is also developed and managed by MoE and KFDA in Korea.

11. Please, pay attention to the references and use the abbreviated journal name when it is in a long way form: references 1, 4, 6, 11, 12, 13, 14, 18, 19, 22, 24, 25, 26, 27, 30, and 31.

Response: Yes, you are correct. We modified the references that you mentioned. Please see the References revised.

- End -

Round  2

Reviewer 1 Report

The authors responded to all comments and the quality of the article was improved. It is my opinion that the work should be accepted.

Reviewer 2 Report

The paper can be accepted after a check of the English